# Time course of visual attention in rats by atomic magnetometer

Fan Liu[1], Zhao Xiang[1], Yuhai Chen[1], Guanzhong Lu[1], Jiahao Wang[1], Jia Yao[2], Ying Zhang[3], Xuejiao Ma[4], Qiang Lin[1]*, Yi Ruan[1]*

1 Laboratory of Quantum Precision Measurement of Zhejiang Province, Center for Optics and Optoelectronics Research, Collaborative Innovation Center for Information Technology in Biological and Medical Physics, College of Science, Zhejiang University of Technology, Hangzhou, China, 2 The First Affiliated Hospital of Zhejiang University School of Medicine, Hangzhou, China, 3 Shaoxing Second Hospital, Shaoxing, China, 4 Beijing Smart-Chip Microelectronics Technology Co., Ltd. Beijing, China

* qlin@zjut.edu.cn (QL); yiruan@zjut.edu.cn (YR)

## Abstract

Atomic magnetometer (AM) is utilized to non-invasively detect event-related magnetic fields (ERMFs) evoked by visual stimuli in rats. The aim of this study was to investigate the relationship between N2-like amplitude and visual attention. To achieve this, we combined the AM with a visual stimulation system and employed the passive single-stimulus paradigm. By measuring the ERMFs at various inter-stimulus intervals (ISIs) with a sensitivity of 20 fT/ $\sqrt{Hz}$, we analyzed the effects of the ISI and the 'habituation' resulting from repeated stimuli on the N2-like amplitude. Our method serves as a valuable reference for studying the passive single-stimulus paradigm and the time course of mammalian attention.

**Data Availability Statement:** All relevant files are available from the Dryad repository (doi:10.5061/dryad.zpc866thf).

**Funding:** This work was supported in part by National Natural Science Foundation of China

## Introduction

Event-related potentials (ERPs) are integral to Electroencephalogram (EEG) research due to their ability to provide a direct measure of brain activity in response to specific events or stimuli. The precise temporal resolution of ERPs allows researchers to map out the sequence of neural processes involved in cognitive functions. These brain responses are typically time-locked to the onset of stimuli, which makes it possible to identify and study distinct ERP components that are associated with different cognitive operations. The ERP waveform is characterized by a series of positive and negative voltage deflections, and each of these components can be linked to specific neural processes [1–3]. These potentials encompass diverse components and are typically distinguished by their polarity and latency [4, 5]. One of the fundamental applications of ERPs is in the oddball paradigm, a well-established experimental protocol used to investigate attention and information processing. In the oddball paradigm, participants are exposed to a series of frequent non-target (standard) stimuli and infrequent target stimuli [5, 6]. The key task for participants is to detect and respond to the target stimuli, which are interspersed randomly among the standard stimuli. This setup creates a contrast that enhances the salience of the target stimuli, making it easier to study the brain's response to novel or important events. Apart from the traditional oddball paradigm, ERPs can also be elicited using

(U20A20219, 61805213); Natural Science Foundation of Zhejiang Province (LQ23H160032). The funders had no role in study design, data collection and analysis, decision to publish, or preparation of the manuscript.

**Competing interests:** The authors have declared that no competing interests exist.

the passive single-stimulus paradigm [7–10]. In this paradigm, the standard stimuli are replaced by the absence of stimuli, and only a single visual or auditory stimulus is presented as the target. This modification allows researchers to examine ERP responses in a context where the participant does not need to perform any specific task or make any discriminations between different stimuli. The passive single-stimulus paradigm is particularly useful for studying brain responses in populations that may find active task participation challenging, such as infants, elderly individuals, or patients with severe cognitive impairments [11]. Research comparing the traditional oddball paradigm with the passive single-stimulus paradigm has yielded intriguing findings. Several studies report no significant differences in the amplitude and latency of ERP components between these two paradigms. This suggests that the brain's response to infrequent stimuli is robust and can be reliably measured even in the absence of a task-related context. However, other studies have indicated that the passive single-stimulus paradigm may produce ERP components with relatively lower amplitude and latency. These differences could be attributed to the reduced attentional demands and lower cognitive load in the passive condition [12, 13].

The N2 (N200) component of visually evoked ERPs is a critical marker in the study of cognitive processes, particularly those related to visual attention and cognitive control. The N2 component typically peaks between 200 milliseconds (ms) and 350 ms after the presentation of a stimulus and is most prominently observed in the anterior scalp regions, particularly over the frontal and central areas [5, 14]. This component is associated with several key cognitive functions, including conflict monitoring, response inhibition, and the processing of novelty and unexpected events. One of the primary factors influencing the amplitude of the N2 component is attention [3, 15]. In visual attention tasks, the amplitude of the N2 component has been found to increase when participants are required to detect, discriminate, or respond to specific visual stimuli. This enhancement of the N2 amplitude under conditions of increased attentional demand suggests that the N2 component is sensitive to the allocation of attentional resources. Despite the well-documented sensitivity of the N2 component to attentional processes, the precise neural mechanisms underlying this modulation remain somewhat elusive. Several hypotheses have been proposed to explain how attention influences the N2 component. One hypothesis is that attentional focus leads to increased neural activation in brain regions responsible for processing the relevant stimuli. This increased activation may result in a larger N2 amplitude. Another possibility is that attention enhances the signal-to-noise ratio (SNR) of the neural responses to relevant stimuli [16]. By suppressing the activity related to irrelevant stimuli, attention could amplify the neural signals associated with the target stimulus, thereby increasing the N2 amplitude. Moreover, top-down modulation, attention may exert top-down control over sensory processing areas, modulating the activity of these regions to prioritize the processing of relevant stimuli. This top-down modulation could enhance the neural response to attended stimuli, reflected in the increased amplitude of the N2 component [17].

ERPs are accompanied by ERMFs, as described by Maxwell's equations. However, EEG signals traverse non-neural tissues such as meninges, cerebrospinal fluid, and skin, which may introduce distortions [18, 19]. Brain magnetic signal measurements offer a promising alternative due to the relatively uniform permeability of brain tissues. High-sensitivity magnetometers, including fluxgate magnetometers, superconducting quantum interference devices (SQUIDs), and AMs, can detect the minute magnetic fields emitted by the brain. While fluxgate magnetometers offer sensitivity of approximately $30\,fT/\sqrt{Hz}$, they are bulky and challenging to integrate [20]. SQUID magnetometers demand cryogenic cooling for optimal performance, resulting in high installation and maintenance costs [21]. AMs, leveraging atom-

light interactions, provide comparable sensitivity to SQUIDs without the need for cryogenics [22, 23].

In our investigation, we explored ERMFs in rats using a custom-built spin-exchange relaxation-free atomic magnetometer (SERF AM). Employing a simplified paradigm, we examined rat responses to single stimuli across various intervals, elucidating the relationship between different time intervals and N2-like response amplitude. Our findings offer valuable insights into the mechanisms underlying visual attention in rats.

## Materials and methods

### Rats preparation

The procedures for the breeding and use of rats were approved by the Ethics Committee of Zhejiang University of Technology (Approval No. 20230703017) and all experimental procedures followed the principles and guidelines for the care and use of animals established by the government and relevant organizations. Balb/c rats, weighing approximately 18 g, were purchased from the Zhejiang Chinese Medical University Laboratory Animal Research Center. The rats were allowed free access to food for more than one week. On the day of the experiment, the rats were fasted for 12 hours and deprived of water for 12 hours prior to the procedure. Before the experiment, the rats were anesthetized with Pentobarbital sodium at a dose of 40 mg/kg, and the experiment was performed 10 minutes after anesthesia. There was no abuse or pain present during the experiment. Eventually, at the end of the experiment, all subjects were euthanised with an intraperitoneal injection of an overdose of sodium pentobarbital.

### Experimental setup

As shown in Fig 1A, the rats were placed on a blister plate to minimize the effects of spontaneous movements. Since the magnetic field strength is linearly related to the distance from the source (i.e., the closer the SERF AM is to the magnetic field source, the larger the detected magnetic field), it was essential to position the SERF AM as close as possible to the rat's brain. However, to avoid recording vibrations caused by the rat's heartbeat, the SERF AM was placed 2 mm away from the brain.

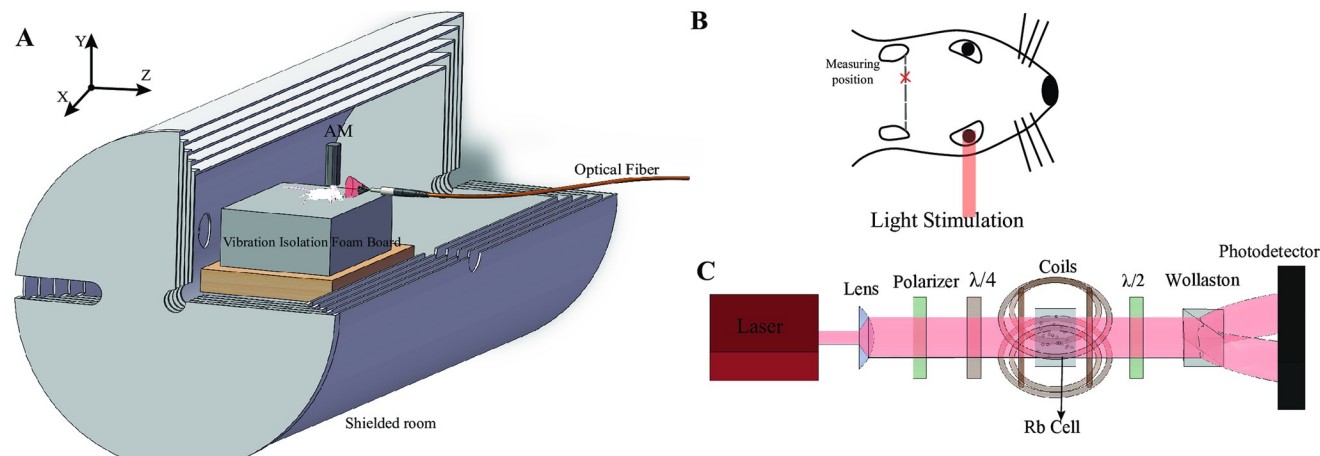

**Fig 1. Configuration of the setup.** A: Five-layer magnetic shield is employed. Both the grating and the silica aerogel pad are made of non-magnetic materials. During the measurement, AM and the rat were placed in the shield, and the measurement direction was on the y-axis. B: Location of visual and auditory stimuli and detection location. C: Internal structure diagram of SERF AM.

During the experiment, both the rats and the SERF AM were placed inside a five-layer Permalloy-shielded barrel with high magnetic permeability to protect them from the geomagnetic field. For visual stimulation, a 510 nm LED was placed outside the shielding barrel [24], and a light-conducting optical fiber introduced light into the barrel at a distance of 1 mm from the rat's right eye. Due to the lateralization effect of the biological brain, the left hemisphere produced the largest amplitude of ERMF when stimulation was applied to the right eye [25, 26]. Considering that the component of interest, N2, exhibited the largest amplitude in the posterior part of the brain, the detection location was situated near the temporal lobe of the left brain region during light stimulation.

## Data processing

1. Dividing the data into blocks, intercepting the data of the stimulated part and the non-stimulated part respectively.

2. Baseline correction of the whole segment data by subtracting the selected baseline data from the entire segment data (200 ms before the start of stimulation was selected in our experiment).

3. Eliminating abnormal data by setting the data range and periodically deleting data beyond the range.

4. Applying a moving average with a window size of 100 ms and a superposition rate of 25%.

5. Averaging the four sample data and drawing the graph.

## Results and discussion

In Fig 1C, the internal structure of the SERF AM probe is depicted. The light is directed through the atomic vapor cell in the z-direction. Prior to entering the vapor cell, the light is blue-shifted by several tens of GHz in relation to the $^{87}$Rb $D_1$ transition $F = 2 \rightarrow F' = 1$. This blue-shifted light is expanded using a plano-convex lens, allowing for more efficient interaction between the light and the atoms. This, in turn, improves the SNR of the system. To manipulate the polarization ellipticity of the light, a linear polarizer and a quarter wave plate are utilized. These optical components enable adjustment of the relative angle between the optical axis of the linear polarizer and the quarter wave plate. Elliptically polarized light can be expressed as a combination of left-circularly polarized light and right-circularly polarized light.

$$E_{in} = E_0 \left( \frac{cos\phi + sin\phi}{\sqrt{2}} \mathscr{L} + \frac{cos\phi + sin\phi}{\sqrt{2}} \mathscr{R} \right), \tag{1}$$

where $E_0$ is incident light intensity. $\phi$ is angle between the optical axis of the linear polarizer and the optical axis of the quarter wave plate. $\mathscr{L}$ is left-circularly polarized light, $\mathscr{L} = e^{\frac{i2\pi vt}{\sqrt{2}}}$. $\mathscr{R}$ is right-circularly polarized light, $\mathscr{R} = e^{\frac{-i2\pi vt}{\sqrt{2}}}$. In our experiment, we set $\phi = \frac{\pi}{8}$. Then the light passes through a $3 \times 3 \times 3$ mm vapor cell containing a drop of enriched $^{87}$Rb atoms and 750 Torr of $N_2$ gas. It was heated to 160°C in the experiment via 1300 kHz AC heating current. Three mutually orthogonal Helmholtz coils were wound outside the vapor cell to cancel the residual magnetic field by using DC current source for power supply which can greatly minimize noise. The compensation currents were automatically determined by a program, which

detects the residual magnetic field by measuring atomic signal. The magnetic field strength generated by the coils can range from -30 nT to 30 nT. When the elliptically polarized light passes through the vapor cell, the atoms in the vapor cell will be pumped. The pumping process can be divided into $\frac{1+s}{2}$ left-circularly photons and $\frac{1-s}{2}$ right-circularly photons pumping together. $\rho(-1/2)$ and $\rho(+1/2)$ are used to describe the electron number density of ground state $m_j = -1/2$ and $m_j = +1/2$, and the change of electronic density of states is

$$\frac{\mathrm{d}}{\mathrm{d}t}\rho(-1/2) \quad = -2R_{op}\rho(-1/2)\frac{1+s}{2} + \frac{1}{2} \times 2R_{op}\rho(-1/2)$$

$$\frac{1+s}{2} + \frac{1}{2} \times 2R_{op}\rho(+1/2)\frac{1-s}{2}, \tag{2}$$

$$\frac{\mathrm{d}}{\mathrm{d}t}\rho(+1/2) \quad = -2R_{op}\rho(+1/2)\frac{1-s}{2} + \frac{1}{2} \times 2R_{op}\rho(+1/2)$$

$$\frac{1-s}{2} + \frac{1}{2} \times 2R_{op}\rho(-1/2)\frac{1+s}{2}, \tag{3}$$

Eq 2 describes the change of ground state $m_j = -1/2$ electron number density. $R_{OP}$ is the rate of absorption of pump light by atoms, which is related to the absorption cross section. $s$ is the average angular momentum of the photon, $s = sin(2\phi)$. $-2R_{op}\rho(+1/2)\frac{1-s}{2}$ is pumping process from ground state $m_j = -1/2$ to excited state under the action of left-circularly light. $\frac{1}{2} \times 2R_{op}\rho(-1/2)\frac{1+s}{2}$ is the process of spontaneous emission of electrons to the ground state $m_j = -1/2$ after the ground state $m_j = -1/2$ electrons are pumped to the excited state by the left-circulary light. $\frac{1}{2} \times 2R_{op}\rho(+1/2)\frac{1-s}{2}$ is the process of spontaneous emission of electrons to the ground state $m_j = -1/2$ after the ground state $m_j = -1/2$ electrons are pumped to the excited state by the right-circulary light. Eq 3 is similar to Eq 2, which describes the change of ground state $m_j = +1/2$ electron number density. According to Eqs 2 and 3, the average angular momentum change of the electron is

$$\frac{\mathrm{d}}{\mathrm{d}t}S_z = \frac{1}{2}R_{OP}(s - 2S_z) - R_{rel}S_z, \tag{4}$$

Since the electron spontaneously depolarizes under the action of many factors, that is, the relaxation process, $R_{rel}$ is added to Eq 4, which is the relaxation rate. After elliptically polarized light passes through the cell, the polarization plane rotates. The rotation angle $\varphi$ is measured by a balanced polarimeter, which consists of a half-wave plate, a Wollaston prism, and a balanced photodetector. Finally, the optoelectronic signal is fed to a lock-in amplifier for demodulation. The rotation angle is due to the different refractive indexes of left-handed circularly polarized light and right-handed circularly polarized light in the polarized vapor cell, which are

$$n_+(\nu) = 1 + \frac{nr_e c^2 f_{D1}}{2\nu}\frac{1+P_{Pr}}{2}\sum_{F,F'}A^{rot}_{F,F'}Im[\mathscr{V}(\nu - \nu_{F,F'})], \tag{5}$$

and

$$n_-(\nu) = 1 + \frac{nr_e c^2 f_{D1}}{2\nu}\frac{1-P_{Pr}}{2}\sum_{F,F'}A^{rot}_{F,F'}Im[\mathscr{V}(\nu - \nu_{F,F'})], \tag{6}$$

respectively. $n$ is total number of ground state atoms, $\nu$ is optical frequency. $f_{D1}$ is $D_1$ line resonance intensity of $^{87}$Rb. $r_e$ is classical radius of electron. $\frac{1+P_{Pr}}{2}$ and $\frac{1-P_{Pr}}{2}$ are the atomic density in

the ground state $m_j = +1/2$ and $m_j = -1/2$ respectively. $Im[\mathscr{V}(v - v_{F,F'})]$ is imaginary part of comprehensive widening, $\mathscr{V}(v - v_0) = \int_0^\infty \mathscr{L}(v - v')\mathscr{G}(v' - v_0)dv'$, where $\mathscr{L}(v - v')$ is collision broadening, Lorentz line, and $\mathscr{G}(v' - v_0)$ is gaussian broadening. $A_{F,F'}^{rot}$ is the normalized relative refractive index intensity between the four hyperfine energy levels of $D_1$ line and is related to the polarization intensity. The phase differences $\varphi_+$ and $\varphi_-$ of the left and right-handed circularly polarized light passing through the atomic vapor cell can be calculated respectively. Therefore, the outgoing light can be expressed as

$$E_{out}(z = l) = E_1\left(\frac{\cos\phi_1 + \sin\phi_1}{\sqrt{2}}e^{i\varphi_+}\mathscr{L}\right. \\ \left. + \frac{\cos\phi_1 - \sin\phi_1}{\sqrt{2}}e^{i\varphi_-}\mathscr{R}\right),$$ (7)

If the change of the outgoing light intensity is ignored, $E_0 = E_1$, and the change of ellipticity of outgoing light, $\phi = \phi_1$, the outgoing light can be expressed as

$$E_{out} = E_{in}e^{i(\varphi_+ + \varphi_-)/2}\begin{vmatrix} e^{i\varphi} \\ e^{-i\varphi} \end{vmatrix},$$ (8)

The outgoing light $E_{out}$ is equivalent to the rotation matrix acting on the incident light $E_{in}$, the rotation angle $\varphi$ is the relative phase difference between the left-handed circularly polarization and the right-handed circularly polarization is

$$\varphi = \frac{\varphi_+ - \varphi_-}{2} = \frac{\pi v l}{c}(n_+ - n_-),$$ (9)

which is equivalent to rotating the incident elliptical light as a whole by an angle $\varphi$. $l$ is the propagation distance. The size of angle $\varphi$ can be calculated by Eqs 5, 6 and 9. After various optimizations of the SERF AM, the sensitivity of the final SERF AM we built is shown in Fig 2A and the bandwidth is shown in Fig 2B.

To investigate the timing of visual attention shifts in rats, we conducted experiments with a stimulation time of 1 s and an ISI ranging from 1 s to 8 s in increments of 1 s [27]. Data from

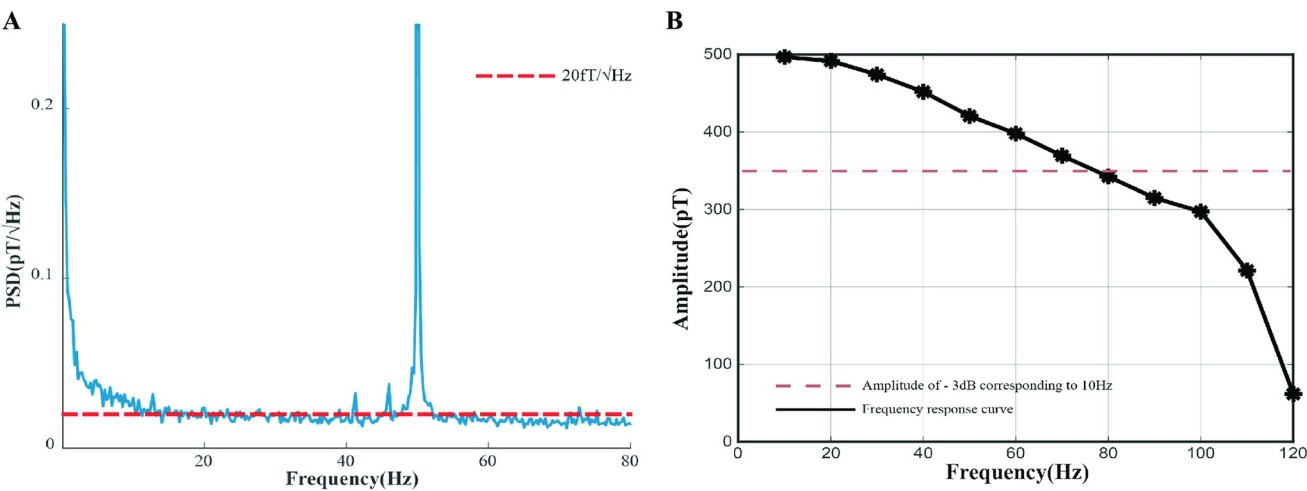

**Fig 2. Optimization results of AM.** A: Sensitivity of SERF AM. B: Frequency response curve with the amplitude beginning at 500 pT.

four samples were recorded. Fig 3A illustrates that regardless of the stimulation interval, a consistently observed negative wave after stimulation was designated as N1, following the naming convention of ERMF, which is independent of cognitive function.

For clarity, we compiled the amplitude of all N1 data in Table 1 (The peak latency is in the supporting information, data in S1 File). It is noteworthy that the amplitude of N1 exhibits changes specifically at ISI values of 7 s and 8 s. To ascertain whether these changes in N1 amplitude at ISIs of 7 s and 8 s are statistically significant, we conducted a statistical analysis, as presented in Fig 4. The results demonstrate a significant variation in N1 amplitude when the ISI is 7 s or 8 s compared to the ISI of 1 s to 5 s.

Given the smaller size and faster signal conduction speed of the rat brain compared to the human brain [28], the N1 component detected in our study may correspond to the N2 component observed in humans, which is associated with visual attention. The earlier ERMF components in rats may be attributed to their low amplitude, which conventional AMs can't detect. Therefore, it is appropriate to refer to it as N2-like. The assignment and latency of this component are linked to the spatial location of the visual stimulus, while its amplitude also varies with different ISIs. We hyposize that the main reason for the observed changes in this component is as follows: within the ISI range of 1–6 s, the amplitude of the N2-like component shows no significant variations. However, in the ISI range of 7–8 s, the amplitude exhibits a sharp increase, $p < 0.05$. This suggests that rats maintain visual attention for approximately 7 s. Within this ISI range, due to sustained attention, the same stimulus enhances adaptability and suppresses the N2-like amplitude. Conversely, when the ISI exceeds 7 s, the attention has dissipated, and upon the arrival of the second visual stimulus, attention is refocused. Consequently, the amplitude of the N2-like component remains unchanged. This ultimately leads to the N2-like amplitude in the 1–6 s ISI range being smaller than that in the 7–8 s ISI range. In a similar situation observed in human auditory sustained stimulation, the N1 amplitude evoked by hearing increases exponentially with longer stimulus intervals, reaching a maximum when intervals exceed 6 s. While the referenced study used the standard paradigm of auditory stimulation to evoke ERMFs, our study employed a single visual stimulus. This method does not require subject cooperation or animal training, making it particularly beneficial for subjects unable to participate in standard paradigms [29].

In order to study the habituation of visual repetitive stimuli in rats, we divided the data into five segments based on the number of repetitive stimuli, namely data from 1–20, 20–40, 40–60, 60–80, and 80–100 repetitive stimuli. Recorded data for four samples. Fig 5 shows that all ISIs show a decreasing trend in amplitude, which is due to habituation after repeated stimuli [30, 31]. However, the amplitude of N2-like does not return to the baseline level after repeated stimuli at ISIs of 7 and 8 s. This may be because the amplitude of N2-like is determined by two parts: firstly, the ISI. When the ISI is greater than 6 s, visual attention refocuses, resulting in a significant increase in amplitude; secondly, the number of repeated stimuli increases, leading to habituation and a decrease in amplitude. Therefore, the amplitude that still exists after repeated stimuli at intervals of 7 and 8 s may be generated by visual attention refocusing.

## Conclusion

In our comprehensive investigation, we conducted meticulous recordings of N2-like amplitude across a range of ISIs utilizing a passive single-stimulus paradigm coupled with an AM. Our rigorous analysis revealed intriguing patterns. Notably, when the ISI extended beyond 6 s, we observed a pronounced and statistically significant augmentation in the N2-like amplitude. However, in EEG studies of auditory and somatosensory inputs in humans, the opposite phenomenon has been observed at shorter ISIs (within 1 s) [32]. Some studies suggest this occurs

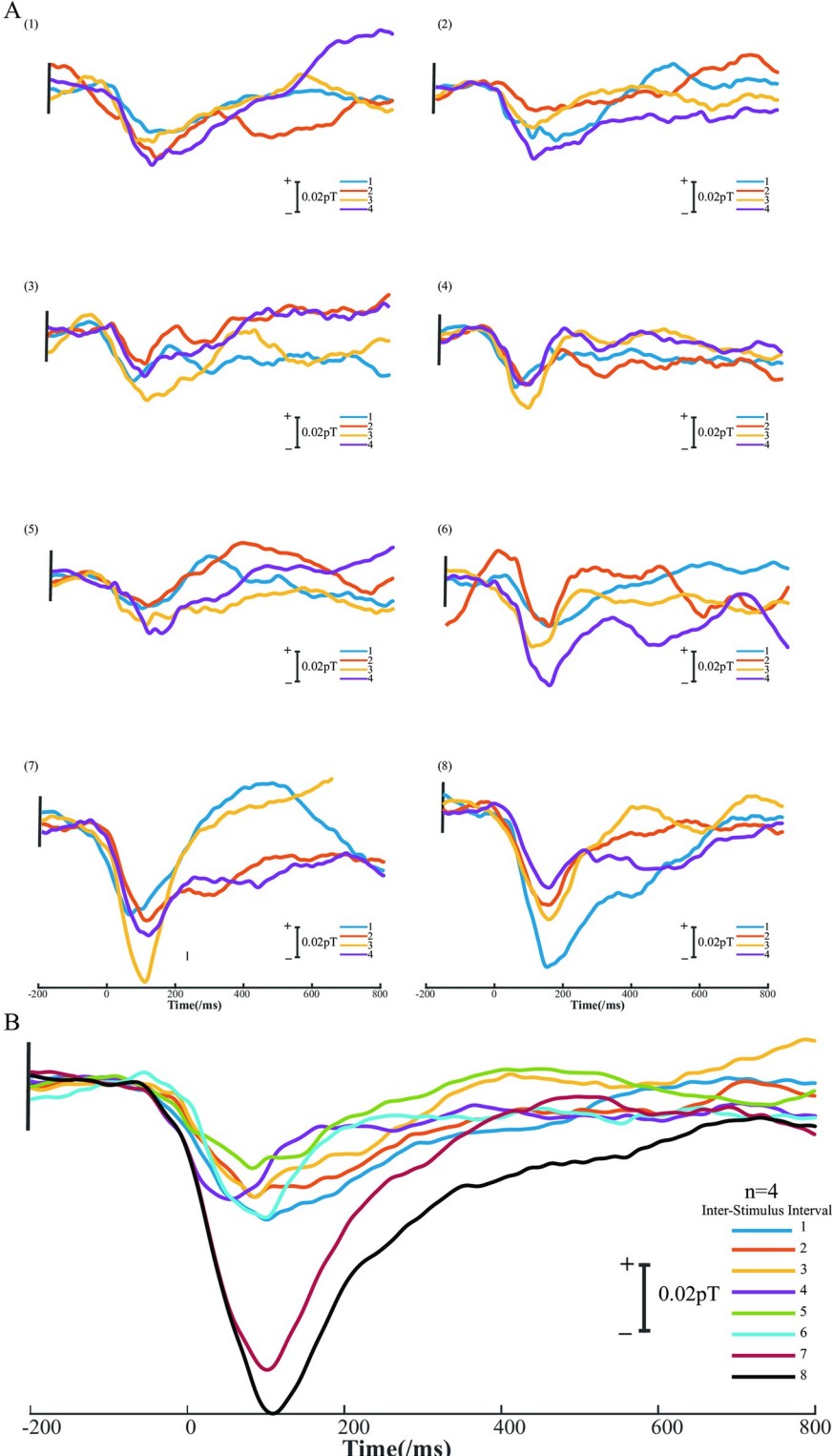

**Fig 3. ERMF with different stimulus intervals.** A: N2-like of stimuli at different time intervals of 1–8 s for four samples. B: Average waveform of N2-like evoked by rats at ISI of 1–8 s, the data of four rats were recorded.

**Table 1. Amplitude and latency of N1 at different ISIs in four rats.**

| Samples | Time interval /s | | | |
|---|---|---|---|---|
|  | 1 | 2 | 3 | 4 |
|  | Amplitude /pT | Amplitude /pT | Amplitude /pT | Amplitude /pT |
| 1 | -0.0303 | -0.0361 | -0.0312 | -0.0359 |
| 2 | -0.0477 | -0.0161 | -0.0228 | -0.0344 |
| 3 | -0.0369 | -0.0275 | -0.0472 | -0.0497 |
| 4 | -0.0525 | -0.0484 | -0.0312 | -0.0342 |
|  | 5 | 6 | 7 | 8 |
|  | Amplitude /pT | Amplitude /pT | Amplitude /pT | Amplitude /pT |
| 1 | -0.0199 | -0.0305 | -0.0947 | -0.1529 |
| 2 | -0.0180 | -0.0304 | -0.0819 | -0.0931 |
| 3 | -0.0307 | -0.0441 | -0.1346 | -0.0766 |
| 4 | -0.0366 | -0.0701 | -0.0769 | -0.107 |

because sensory input generates a large excitatory response in the neuronal population responsible for the N1 wave. As this response propagates through association fibers, it triggers a less precisely time-locked secondary excitatory response in neighboring inhibitory interneurons, leading to sustained inhibition of the N1-generating neuronal population. Since the inhibitory feedback takes time to develop, a second stimulus arriving before full inhibition results in a larger response than one arriving after inhibition has occurred [33, 34]. However, this model does not apply for longer ISIs. We therefore proposed a novel hypothesis that a rat's attentional capacity may approximate a temporal window of approximately 6 s. It indicates that with

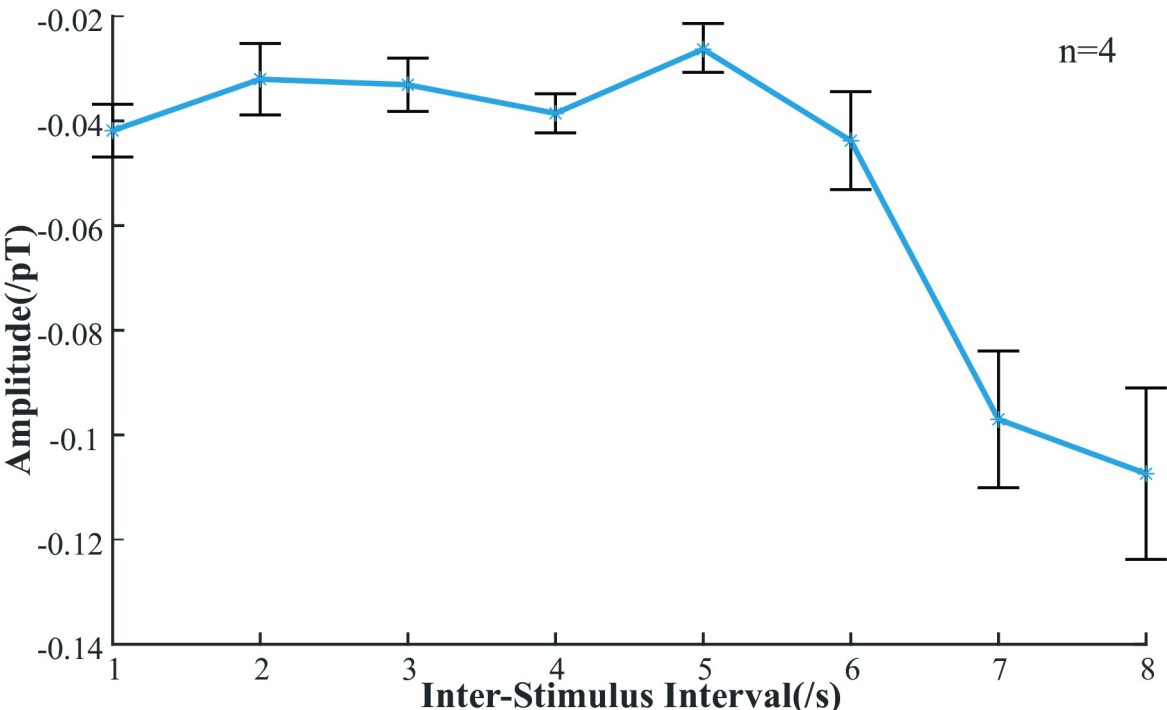

**Fig 4. The change curve of average ERMF for four samples with different ISIs.**

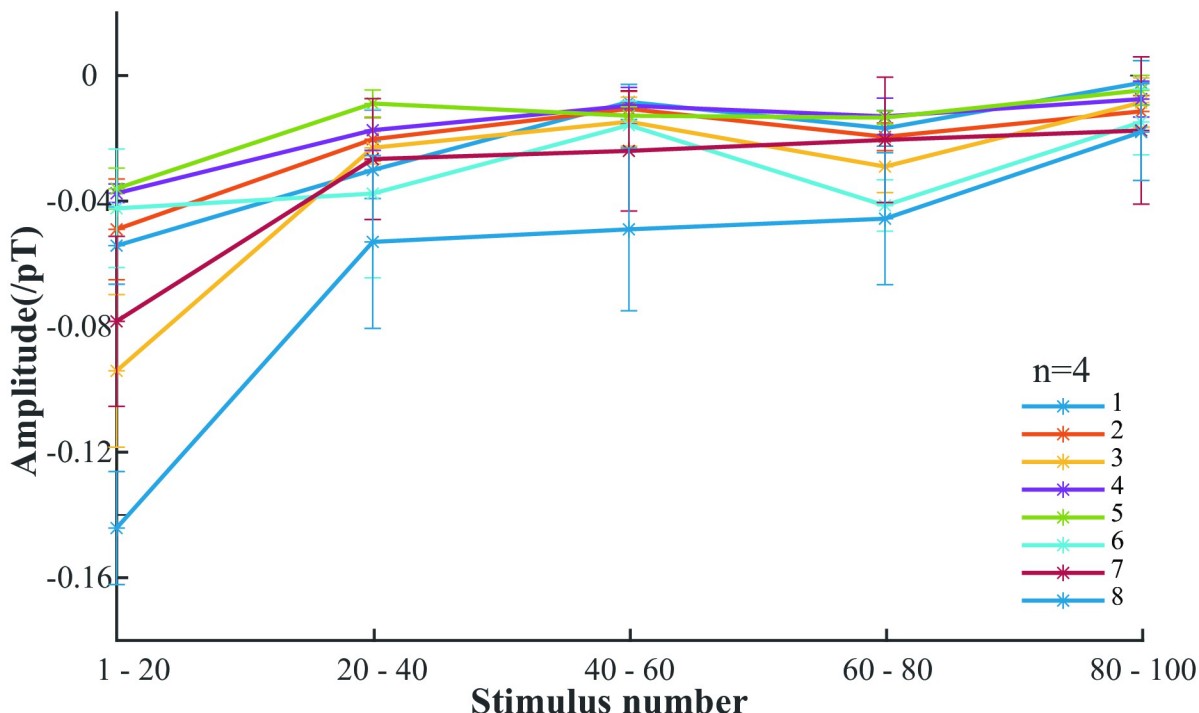

**Fig 5. The change curve of the average ERMF of four samples under repeated stimuli with different ISIs.**

longer ISIs, the N2-like amplitude exhibits a heightened response, indicative of a sustained attentional state. Conversely, when the ISI was truncated to durations shorter than 6 s, we noted a rapid reorientation of attention, resulting in a conspicuous absence of N2-like amplitude suppression. The observed patterns of N2-like amplitude modulation highlight the importance of temporal dynamics in attentional processing. The ability to sustain attention over longer intervals, as indicated by the increased N2-like amplitude at ISIs beyond 6 s, suggests a temporal threshold for sustained attentional engagement. This has implications for designing experiments and interpreting data related to attentional capacity and cognitive control.

Our investigation also delved into the dynamics of stimulus repetition and its impact on the N2-like amplitude. Remarkably, as the number of stimulus repetitions increased, we observed a gradual diminishment in the N2-like amplitude across varying ISIs. This intriguing observation underscores the influential role of habituation, where in repeated exposure to stimuli engenders a diminishing neural response [35, 36]. The reduction in neural response to repeated stimuli reflects the brain's ability to adapt to familiar stimuli, which is a fundamental aspect of learning and memory. Notably, however, when the ISI was shorter than 6 s, the N2-like amplitude tended to converge towards baseline levels. In contrast, for ISIs surpassing the 6-s threshold, the N2-like amplitude exhibited a remarkable decline to approximately 0.04 pT. This nuanced exploration suggests that the amplitude of the N2-like component is shaped by two primary factors: the temporal characteristics of the stimulus interval and the cumulative effects of stimulus repetition. The findings imply that attentional mechanisms and habituation play crucial roles in modulating neural responses to repetitive stimuli. Understanding these dynamics can provide deeper insights into the neural basis of attention and cognitive processing in both animal models and potentially in human studies.

Our approach entailed the utilization of an AM, a sophisticated tool enabling the non-invasive detection of ERMFs elicited by visual stimuli in rats. By seamlessly integrating AM with a visual stimulation system and employing the passive single-stimulus paradigm, we meticulously measured ERMF at varying ISIs with a sensitivity of 20 fT/$\sqrt{\text{Hz}}$. Subsequent analysis meticulously scrutinized the interplay between habituation effects induced by ISIs and the modulatory influence of stimulus repetition on N2-like amplitude.

Collectively, our findings offer valuable insights for advancing the clinical application of a single stimulus paradigm and chart a new trajectory for investigating the temporal dynamics of visual attention in small mammalian models. (A coil simulation of the AM is included in the Supplementary file along with the amplitude and latency of N1, data in S1 File).

## Supporting information

**S1 File. Magnetic field distribution of the external magnetic field coils; the amplitude and latency of ERMF for each group of samples.**
(DOCX)

## Author Contributions

**Conceptualization:** Guanzhong Lu.

**Data curation:** Yuhai Chen, Xuejiao Ma.

**Formal analysis:** Jiahao Wang.

**Funding acquisition:** Qiang Lin.

**Investigation:** Zhao Xiang, Yuhai Chen.

**Methodology:** Fan Liu.

**Project administration:** Qiang Lin.

**Resources:** Jia Yao, Ying Zhang, Yi Ruan.

**Software:** Zhao Xiang.

**Writing – original draft:** Fan Liu.

**Writing – review & editing:** Yi Ruan.

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
