## [Decision Letter · Decision Letter 0]

7 Jun 2024

PONE-D-24-14635Time course of visual attention in rats by atomic magnetometerPLOS ONE

Dear Dr. Ruan,

Thank you for submitting your manuscript to PLOS ONE. After careful consideration, we feel that it has merit but does not fully meet PLOS ONE’s publication criteria as it currently stands. Therefore, we invite you to submit a revised version of the manuscript that addresses the points raised during the review process.

We look forward to receiving your revised manuscript.

Kind regards,

Mingsen Deng

Academic Editor

PLOS ONE

“This work was supported in part by National Natural Science Foundation of China (U20A20219, 61805213); Natural Science Foundation of Zhejiang Province (LQ23H160032).”

5. We note that Figure 1 in your submission contain copyrighted images. All PLOS content is published under the Creative Commons Attribution License (CC BY 4.0), which means that the manuscript, images, and Supporting Information files will be freely available online, and any third party is permitted to access, download, copy, distribute, and use these materials in any way, even commercially, with proper attribution. For more information, see our copyright guidelines: http://journals.plos.org/plosone/s/licenses-and-copyright.

6. Please include a separate caption for each figure in your manuscript.

Reviewers' comments:

Reviewer's Responses to Questions

**Comments to the Author**

1. Is the manuscript technically sound, and do the data support the conclusions?

Reviewer #1: Partly

Reviewer #2: Yes

2. Has the statistical analysis been performed appropriately and rigorously? 

Reviewer #1: Yes

Reviewer #2: Yes

3. Have the authors made all data underlying the findings in their manuscript fully available?

Reviewer #1: Yes

Reviewer #2: Yes

4. Is the manuscript presented in an intelligible fashion and written in standard English?

Reviewer #1: Yes

Reviewer #2: Yes

5. Review Comments to the Author

Reviewer #1: 1.The design method is not introduced in detail, and it feels like a manuscript in the form of a letter rather than a research article.Therefore, the paper needs to give a design method in detail .

2.In the result part, there is no data comparison with recent methods, so the innovation of the method cannot be verified. Only the Amplitude and latency of N1 at different ISIs in four rats are given. The paper needs to supplement the data comparison of relevant methods.

Reviewer #2: This manuscript draft outlines a research study examining the dynamics of visual attention in rats using an atomic magnetometer. The study investigates the relationship between N2-like amplitude and visual attention.

The goal is to measure the N2-like component's amplitude in response to visual stimuli at different inter-stimulus intervals (ISIs) using a passive single-stimulus paradigm. Key findings include:

- The N2-like component's amplitude significantly increases when the ISI exceeds 6 seconds, indicating a sustained attentional state in rats.

- The amplitude decreases with repeated stimuli, showing habituation effects.

- At ISIs of 7 and 8 seconds, the amplitude does not return to baseline levels, suggesting that visual attention refocusing may help maintain the amplitude.

I agree that the study provides valuable insights into the time course of mammalian attention and serves as a reference for future research using the passive single-stimulus paradigm.

The authors should consider rephrasing terms like "intriguing" and "remarkable" in the conclusion section to maintain scientific objectivity and neutrality.

6. PLOS authors have the option to publish the peer review history of their article (what does this mean?). If published, this will include your full peer review and any attached files.

Reviewer #1: No

Reviewer #2: **Yes: **Prof. Dr. Murat Özgören, MD PhD

---

## [Author Response · Author response to Decision Letter 0]

1 Aug 2024

Reviewer #1: 

Question 1: The design method is not introduced in detail, and it feels like a manuscript in the form of a letter rather than a research article. Therefore, the paper needs to give a design method in detail .

Response 1: Thanks for the reviewer’s suggestions. We have added details of our designed method, introduction and conclusion. At page1-2, line 2-54, page 3, line 73-95, page 9, line 213-243.

Question 2: In the result part, there is no data comparison with recent methods, so the innovation of the method cannot be verified. Only the Amplitude and latency of N1 at different ISIs in four rats are given. The paper needs to supplement the data comparison of relevant methods.

Response 2: Thanks for the reviewer’s suggestions. We add the comparison with other methods. At page 1, line 2 , page 3, line 73, page 6, line 167, and page 9, line 213. 

Reviewer #2: 

Comments: This manuscript draft outlines a research study examining the dynamics of visual attention in rats using an atomic magnetometer. The study investigates the relationship between N2-like amplitude and visual attention.

The goal is to measure the N2-like component's amplitude in response to visual stimuli at different inter-stimulus intervals (ISIs) using a passive single-stimulus paradigm. Key findings include:

- The N2-like component's amplitude significantly increases when the ISI exceeds 6 seconds, indicating a sustained attentional state in rats.

- The amplitude decreases with repeated stimuli, showing habituation effects.

- At ISIs of 7 and 8 seconds, the amplitude does not return to baseline levels, suggesting that visual attention refocusing may help maintain the amplitude.

I agree that the study provides valuable insights into the time course of mammalian attention and serves as a reference for future research using the passive single-stimulus paradigm.

Response: We thank Prof. Dr. Murat Özgören for his assessment of our manuscript. The study provides valuable insights into the time course of mammalian attention, specifically in rats, using a passive single-stimulus paradigm. The findings demonstrate that the N2-like component amplitude significantly increases with ISIs exceeding 6 seconds, suggesting a sustained attentional state. The observed decrease in amplitude with repeated stimuli indicates habituation effects. Additionally, the maintenance of amplitude at ISIs of 7 and 8 seconds implies that visual attention refocusing may contribute to sustaining the response amplitude. These results enhance our understanding of visual attention dynamics and offer a reference point for future research in this area.

We agree with your suggestion that the terms "intellectualizing" and "remarkable" are inappropriate and have already changed them.

---

## [Decision Letter · Decision Letter 1]

8 Sep 2024

PONE-D-24-14635R1Time course of visual attention in rats by atomic magnetometer：A study based on event-related magnetic fieldsPLOS ONE

Dear Dr. Ruan,

Thank you for submitting your manuscript to PLOS ONE. After careful consideration, we feel that it has merit but does not fully meet PLOS ONE’s publication criteria as it currently stands. Therefore, we invite you to submit a revised version of the manuscript that addresses the points raised during the review process.

We look forward to receiving your revised manuscript.

Kind regards,

Mingsen Deng

Academic Editor

PLOS ONE

Journal Requirements:

Reviewers' comments:

Reviewer's Responses to Questions

**Comments to the Author**

1. If the authors have adequately addressed your comments raised in a previous round of review and you feel that this manuscript is now acceptable for publication, you may indicate that here to bypass the “Comments to the Author” section, enter your conflict of interest statement in the “Confidential to Editor” section, and submit your "Accept" recommendation.

Reviewer #1: All comments have been addressed

Reviewer #2: All comments have been addressed

2. Is the manuscript technically sound, and do the data support the conclusions?

Reviewer #1: Yes

Reviewer #2: Yes

3. Has the statistical analysis been performed appropriately and rigorously? 

Reviewer #1: Yes

Reviewer #2: Yes

4. Have the authors made all data underlying the findings in their manuscript fully available?

Reviewer #1: Yes

Reviewer #2: Yes

5. Is the manuscript presented in an intelligible fashion and written in standard English?

Reviewer #1: Yes

Reviewer #2: Yes

6. Review Comments to the Author

Reviewer #1: The manuscript addresses an interesting issue on event-related magnetic fields, and the research question is clearly stated. I have several suggestions that I believe will enhance the quality of the manuscript :

(1) In the result section, It is suggested that authors add more relevant literature from the last five years for comparative analysis to establish the novelty and significance of the work.

(2) The manuscript is well-written, but some small mistakes can benefit form improvement. For example, there are two extra question marks in the sentence line 27 ( impairments??. )

Reviewer #2: The resubmission has improved the manuscript. The authors have responded to the reviewers comments acccordingly.

7. PLOS authors have the option to publish the peer review history of their article (what does this mean?). If published, this will include your full peer review and any attached files.

Reviewer #1: No

Reviewer #2: **Yes: **Prof. Dr. MD PhD MURAT ÖZGÖREN

---

## [Author Response · Author response to Decision Letter 1]

23 Sep 2024

Reviewer #1: 

Question 1: In the result section, It is suggested that authors add more relevant literature from the last five years for comparative analysis to establish the novelty and significance of the work.

Response 1: Thank you for your comment, we have added a comparison with other articles.

Question 2: The manuscript is well-written, but some small mistakes can benefit form improvement. For example, there are two extra question marks in the sentence line 27 ( impairments??. )

Response 2: Thank you for your comments, we have revised it and checked the full text.

Reviewer #2: 

Response: We thank Professor Murat Özgören for his approval of our paper. This study provides valuable insights into the time course of mammalian attention, particularly in rats, using a passive single-stimulus paradigm. The findings showed that the N2-like component amplitude increased significantly when the ISI exceeded 6 seconds, indicating a sustained state of attention. The decrease in amplitude observed upon repeated stimulation suggests the presence of a habituation effect. Furthermore, the maintenance of amplitude at ISIs of 7 and 8 s implies that refocusing of visual attention may contribute to the maintenance of response amplitude. These results enhance our understanding of the dynamics of visual attention and provide a reference point for future research in this area.

---

## [Decision Letter · Decision Letter 2]

10 Oct 2024

Time course of visual attention in rats by atomic magnetometer：A study based on event-related magnetic fields

PONE-D-24-14635R2

Dear Dr. Ruan,

We’re pleased to inform you that your manuscript has been judged scientifically suitable for publication and will be formally accepted for publication once it meets all outstanding technical requirements.

Kind regards,

Mingsen Deng

Academic Editor

PLOS ONE

Additional Editor Comments (optional):

Reviewers' comments:

Reviewer's Responses to Questions

**Comments to the Author**

1. If the authors have adequately addressed your comments raised in a previous round of review and you feel that this manuscript is now acceptable for publication, you may indicate that here to bypass the “Comments to the Author” section, enter your conflict of interest statement in the “Confidential to Editor” section, and submit your "Accept" recommendation.

Reviewer #1: All comments have been addressed

2. Is the manuscript technically sound, and do the data support the conclusions?

Reviewer #1: Yes

3. Has the statistical analysis been performed appropriately and rigorously? 

Reviewer #1: Yes

4. Have the authors made all data underlying the findings in their manuscript fully available?

Reviewer #1: Yes

5. Is the manuscript presented in an intelligible fashion and written in standard English?

Reviewer #1: Yes

6. Review Comments to the Author

Reviewer #1: The paper presents a time course of visual attention in rats by atomic magnetometer and all comments have been addressed in this submitted manuscript.

7. PLOS authors have the option to publish the peer review history of their article (what does this mean?). If published, this will include your full peer review and any attached files.

Reviewer #1: No

---

## [Editor Report · Acceptance letter]

18 Oct 2024

PONE-D-24-14635R2 

PLOS ONE

Dear Dr. Ruan, 

I'm pleased to inform you that your manuscript has been deemed suitable for publication in PLOS ONE. Congratulations! Your manuscript is now being handed over to our production team.

Kind regards, 

on behalf of

Dr. Mingsen Deng 

Academic Editor

PLOS ONE